# DINO-WM: World Models on Pre-trained Visual Features enable Zero-shot Planning

**Gaoyue Zhou**[1]  **Hengkai Pan**[1]  **Yann LeCun**[1 2]  **Lerrel Pinto**[1]

## Abstract

The ability to predict future outcomes given control actions is fundamental for physical reasoning. However, such predictive models, often called world models, remain challenging to learn and are typically developed for task-specific solutions with online policy learning. To unlock world models' true potential, we argue that they should 1) be trainable on offline, pre-collected trajectories, 2) support test-time behavior optimization, and 3) facilitate task-agnostic reasoning. To this end, we present DINO World Model (DINO-WM), a new method to model visual dynamics without reconstructing the visual world. DINO-WM leverages spatial patch features pre-trained with DINOv2, enabling it to learn from offline behavioral trajectories by predicting future patch features. This allows DINO-WM to achieve observational goals through action sequence optimization, facilitating task-agnostic planning by treating goal features as prediction targets. We demonstrate that DINO-WM achieves zero-shot behavioral solutions at test time on six environments without expert demonstrations, reward modeling, or pre-learned inverse models, outperforming prior state-of-the-art work across diverse task families such as arbitrarily configured mazes, push manipulation with varied object shapes, and multi-particle scenarios.

## 1. Introduction

Robotics and embodied AI have seen tremendous progress in recent years. Advances in imitation learning and reinforcement learning have enabled agents to learn complex behaviors across diverse tasks (Agarwal et al., 2022; Zhao et al., 2023; Lee et al., 2024; Ma et al., 2024; Hafner et al., 2024; Hansen et al., 2024; Haldar et al., 2024; Jia et al., 2024). Despite this progress, generalization remains a major challenge (Zhou et al., 2023). Existing approaches predominantly rely on policies that, once trained, operate in a feed-forward manner during deployment—mapping observations to actions without any further optimization or reasoning. Under this framework, successful generalization inherently requires agents to possess solutions to all possible tasks and scenarios once training is complete, which is only possible if the agent has seen similar scenarios during training (Reed et al., 2022; Brohan et al., 2023b;a; Etukuru et al., 2024). However, it is neither feasible nor efficient to learn solutions for all potential tasks and environments in advance.

Instead of learning the solutions to all possible tasks during training, an alternative is to fit a dynamics model on training data and optimize task-specific behavior at runtime. These dynamics models, also called world models (Ha & Schmidhuber, 2018), have a long history in robotics and control (Sutton, 1991; Todorov & Li, 2005; Williams et al., 2017). More recently, several works have shown that world models can be trained on raw sensory data (Hafner et al., 2019; Micheli et al., 2023; Robine et al., 2023; Hansen et al., 2024; Hafner et al., 2024). This enables flexible use of model-based optimization to obtain policies as it circumvents the need for explicit state-estimation. Despite this, significant challenges remain in its use for solving general-purpose tasks.

To understand the challenges in world modeling, let us consider the two broad paradigms in learning world models: online and offline. In the online setting, access to the environment is often required so data can be continuously collected to improve the world model, which in turn improves the policy and the subsequent data collection. However, the online world model is only accurate in the cover of the policy that was being optimized. Hence, while it can be used to train powerful task-specific policies, it requires retraining for every new task even in the same environment. Instead, in the offline setting, the world model is trained on an offline dataset of collected trajectories in the environment, which removes its dependence on the task specificity given sufficient coverage in the dataset. However, when

[1]Courant Institute, New York University [2]Meta AI. Correspondence to: Gaoyue Zhou <gz2123@nyu.edu>.

*Proceedings of the 42nd International Conference on Machine Learning*, Vancouver, Canada. PMLR 267, 2025. Copyright 2025 by the author(s).

required to solve a task, methods in this domain require strong auxiliary information which can take the form of expert demonstrations (Pathak et al., 2018; Wang et al., 2023), structured keypoints (Ko et al., 2023; Wen et al., 2024), access to pretrained inverse models (Du et al., 2023; Ko et al., 2023) or dense reward functions (Ding et al., 2024), all of which reduce the generality of using offline world models. The central question in building better offline world models is if there is alternate auxiliary information that does not compromise its generality?

In this work, we present DINO-WM, a new and simple method to build task-agnostic world models from an offline dataset of trajectories (Figure 1). DINO-WM models the world dynamics on compact embeddings of the world, rather than the raw observations themselves. For the embedding, we use pretrained patch-features from the DINOv2 model, which provides both a spatial and object-centric representation prior. We conjecture that this pretrained representation enables robust and consistent world modeling, which relaxes the necessity for task-specific data coverage. Given these visual embeddings and actions, DINO-WM uses the ViT architecture to predict future embeddings. Once this model is trained on the offline dataset, planning to solve tasks is constructed as visual goal reaching, i.e. to reach a future desired goal given the current observation. Since the predictions by DINO-WM are high quality (see Figure 4), we can simply use model predictive control with inference-time optimization to reach desired goals without any extra information during testing.

DINO-WM is experimentally evaluated on six environment suites spanning maze navigation, sliding manipulation, robotic arm control, and deformable object manipulation tasks. Our experiments reveal the following findings:

- DINO-WM produces high-quality future world modeling that can be measured by improved visual reconstruction from trained decoders. On LPIPS metrics for our hardest tasks, this improves upon prior state-of-the-art work by 56% (See Section 4.7).

- Given the latent world models trained using DINO-WM, we show high success for reaching arbitrary goals on our hardest tasks, improving upon prior work by 45% on average (See Section 4.3).

- DINO-WM can be trained across environment variations within a task family (e.g. different maze layouts for navigation or different object shapes for manipulation) and achieve higher rates of success compared to prior work (See Section 4.5).

Code and models for DINO-WM are open-sourced to ensure reproducibility and videos of planning are made available on our project website: `https://dino-wm.github.io/`.

## 2. Related Work

We build on top of several works in developing world models, optimizing behaviors from them, and leveraging compact visual representations. For conciseness, we only discuss the ones most relevant to DINO-WM.

**Model-based Learning:** Learning from models of dynamics has a rich literature spanning the fields of control, planning, and robotics (Sutton, 1991; Todorov & Li, 2005; Astolfi et al., 2008; Holkar & Waghmare, 2010; Williams et al., 2017). Recent works have shown that modeling dynamics and predicting future states can significantly enhance vision-based learning for embodied agents across various applications, including online reinforcement learning (Micheli et al., 2023; Robine et al., 2023; Hansen et al., 2024; Hafner et al., 2024), exploration (Sekar et al., 2020; Mendonca et al., 2021; 2023a), planning (Watter et al., 2015) (Finn & Levine, 2017; Ebert et al., 2018; Hafner et al., 2019), and imitation learning (Pathak et al., 2018). Several of these approaches initially focused on state-space dynamics (Deisenroth & Rasmussen, 2011; Lenz et al., 2015; Chua et al., 2018; Nagabandi et al., 2019), and have since been extended to handle image-based inputs, which we address in this work. These world models can predict future states in either pixel space (Finn & Levine, 2017; Ebert et al., 2018; Ko et al., 2023; Du et al., 2023) or latent representation space (Yan et al., 2021). Predicting in pixel space, however, is computationally expensive due to the need for image reconstruction and the overhead of using diffusion models (Ko et al., 2023). On the other hand, latent-space prediction is typically tied to image reconstruction objectives (Hafner et al., 2019; Micheli et al., 2023; Hafner et al., 2024), which raises concerns about whether the learned features contain sufficient information about the task. Moreover, many of these models incorporate reward prediction (Micheli et al., 2023; Robine et al., 2023; Hafner et al., 2024), or use reward prediction as an auxiliary objective to learn the latent representation (Hansen et al., 2022; 2024), inherently making the world model task-specific. In this work, we aim to decouple task-dependent information from latent-space prediction, striving to develop a versatile and task-agnostic world model capable of generalizing across different scenarios.

**Generative Models as World Models:** With the recent excitement of large scale foundation models, there have been initiatives on building large-scale video generation world models conditioned on agent's actions in the domain of self-driving (Hu et al., 2023), control (Yang et al., 2023; Bruce et al., 2024), and general-purpose video generation (Liu et al., 2024). These models aim to generate video predictions conditioned on text or high-level action

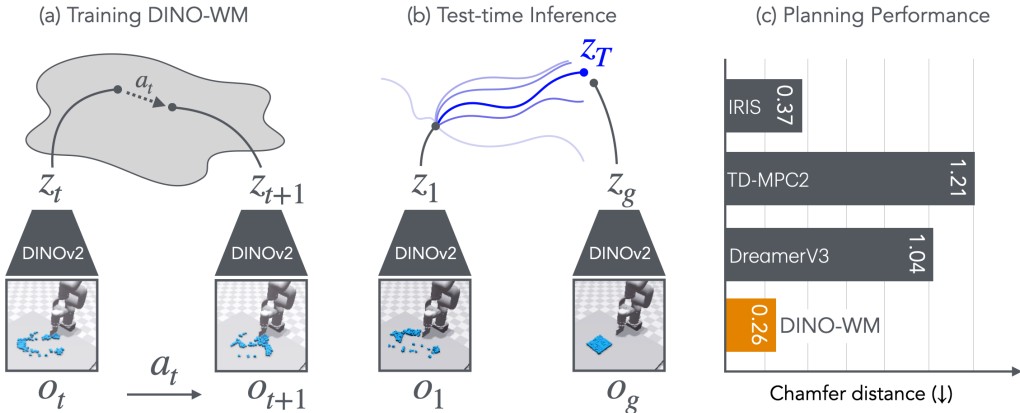

*Figure 1.* We present DINO-WM, a method for training visual models by using pretrained DINOv2 embeddings of image frames (a). Once trained, given a target observation $o_T$, we can directly optimize agent behavior by planning through DINO-WM using model predictive control (b). The use of pretrained embeddings significantly improves performance over prior state-of-the-art world models (c).

sequences. While these models have demonstrated utility in downstream tasks like data augmentations, their reliance on language conditioning limits their application when precise visually indicative goals need to be reached. Additionally, the use of diffusion models for video generation makes them computationally expensive, further restricting their applicability for test-time optimization techniques such as MPC. In this work, we aim to build a world model in latent space instead of raw pixel space, enabling more precise planning and control.

**Pretrained Visual Representations:** Significant advancements have been made in the field of visual representation learning, where compact features that capture spatial and semantic information can be readily used for downstream tasks. Pre-trained models like ImageNet pre-trained ResNet (He et al., 2016), I-JEPA (Assran et al., 2023), and DINO (Caron et al., 2021; Oquab et al., 2024) for images, as well as V-JEPA (Bardes et al., 2024) for videos, and R3M (Nair et al., 2022), MVP (Xiao et al., 2022) for robotics have allowed fast adaptation to downstream tasks as they contain rich spatial and semantic information. While many of these models represent images using a single global feature, the introduction of Vision Transformers (ViTs) (Dosovitskiy et al., 2021) has enabled the use of pre-trained patch features, as demonstrated by DINO (Caron et al., 2021; Oquab et al., 2024). DINO employs a self-distillation loss that allows the model to learn representations effectively, capturing semantic layouts and improving spatial understanding within images. In this work, we leverage DINOv2's patch embeddings to train our world model, and demonstrate that it serves as a versatile encoder capable of handling various precise tasks.

## 3. DINO World Models

**Overview and Problem Formulation:** Our work follows the vision-based control task framework, which models the environment as a partially observable Markov decision process (POMDP). The POMDP is defined by the tuple $(\mathcal{O}, \mathcal{A}, p)$, where $\mathcal{O}$ represents the observation space, and $\mathcal{A}$ denotes the action space. The dynamics of the environment are modeled by the transition distribution $p(o_{t+1} \mid o_{\leq t}, a_{\leq t})$, which predicts future observations based on past actions and observations.

In this work, we aim to learn task-agnostic world models from precollected offline datasets, and use these world models to perform visual reasoning and control at test time. At test time, our system starts from an arbitrary environment state and is provided with a goal observation in the form of an RGB image, in line with prior works (Ebert et al., 2018; Wu et al., 2020; Mendonca et al., 2023b), and is asked to perform a sequence of actions $a_0, ..., a_T$ to reach the goal state. This approach differs from the world models used in online reinforcement learning (RL) where the objective is to optimize the rewards for a fixed set of tasks at hand (Hafner et al., 2024; Hansen et al., 2024), or from text-conditioned world models, where the goals are specified through text prompts (Du et al., 2023; Ko et al., 2023).

### 3.1. DINO-based World Models (DINO-WM)

We model the dynamics of the environment in the latent space. More specifically, at each time step $t$, our world model consists of the following components:

$$\text{Observation model:} \quad z_t \sim \text{enc}_\theta(z_t \mid o_t)$$
$$\text{Transition model:} \quad z_{t+1} \sim p_\theta(z_{t+1} \mid z_{t-H:t}, a_{t-H:t})$$
$$\text{Decoder model:} \quad \hat{o}_t \sim q_\theta(o_t \mid z_t)$$

(optional for visualization)

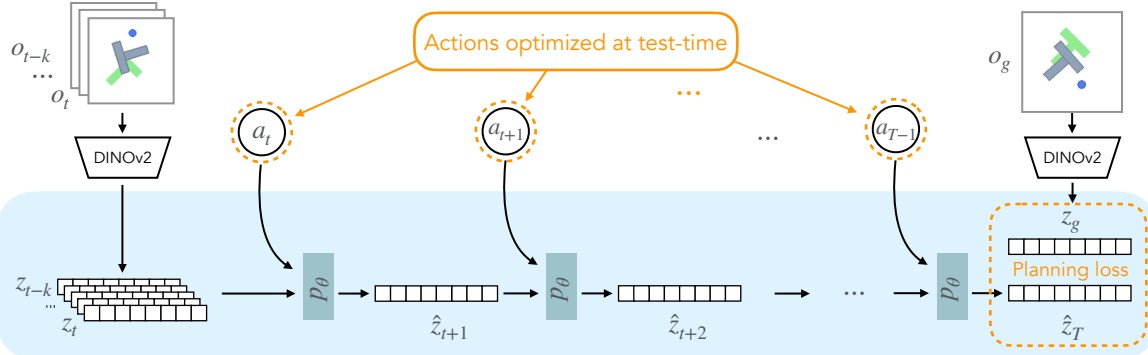

*Figure 2.* Architecture of DINO-WM. Given observations $o_{t-k:t}$, we optimize the sequence of actions $a_{t:T-1}$ to minimize the predicted loss to the desired goal $o_g$. All forward computation is done in the latent space $z$. Here $p_\theta$ indicates DINO-WM's dynamics model, which is used for making future predictions.

where the observation model encodes image observations to latent states $z_t$, and the transition model takes in a history of past latent states of length $H$. The decoder model takes in a latent $z_t$, and reconstructs the image observation $o_t$. We use $\theta$ to denote the parameters of these models. Note that our decoder is entirely optional, as the training objectives for the decoder are independent for training the rest part of the world model. This eliminates the need to reconstruct images both during training and testing, which reduces computational costs compared to otherwise coupling together the training of the observational model and the decoder, as in (Micheli et al., 2023; Hafner et al., 2024). We ablate and show the effectiveness of this choice in Appendix A.4.2.

DINO-WM models only the information available from offline trajectory data in an environment, in contrast to recent online RL world models that also require task-relevant information, such as rewards (Hafner et al., 2020; Hansen et al., 2022; 2024), discount factors (Hafner et al., 2022; Robine et al., 2023), and termination conditions (Micheli et al., 2023; Hafner et al., 2024).

### 3.1.1. OBSERVATION MODEL

To learn a generic world model across many environments and the real world, we argue that the observation model should 1) be task and environment independent, and 2) capture rich spatial information for navigation and manipulation. Contrary to previous work where the observation model is always learned for the task at hand (Hafner et al., 2024), we argue instead that it can be inefficient and often not possible to learn a good observation model from scratch when facing a new environment, as perception is a general task that benefits from large-scale internet data. Therefore, we use the pre-trained DINOv2 model as our world model's observation model, leveraging its strong spatial understanding for tasks like object detection, semantic segmentation, and depth estimation (Oquab et al., 2024). The observation model remains frozen during training and testing. At

each time step $t$, it encodes an image $o_t$ to patch embeddings $z_t \in \mathbb{R}^{N \times E}$, where $N$ denotes the number of patches, and $E$ denotes the embedding dimension. This process is visualized in Figure 2.

### 3.1.2. TRANSITION MODEL

We adopt the ViT architecture (Dosovitskiy et al., 2021) for the transition model due to its suitability for processing patch features. We remove the tokenization layer, as it operates on patch embeddings, effectively transforming it into a decoder-only transformer. We further make a few modifications to the architecture to allow for additional conditioning on proprioception and controller actions.

Our transition model takes in a history of past latent states $z_{t-H:t-1}$ and actions $a_{t-H:t-1}$, where $H$ is a hyperparameter denoting the context length of the model, and predicts the latent state at next time step $z_t$. To properly capture the temporal dependencies, where the world state at time $t$ should only depend on previous observations and actions, we implement a causal attention mechanism in the ViT model, enabling the model to predict latents autoregressively at a frame level. Specifically, each patch vector $z_t^i$ for the latent state $z_t$ attends to $\{z_{t-H:t-1}^i\}_{i=1}^N$. This is different from past work IRIS (Micheli et al., 2023) which similarly represents each observation as a sequence of vectors, but autoregressively predict $z_t^i$ at a token level, attending to $\{z_{t-H:t-1}^i\}_{i=1}^N$ as well as $\{z_t^i\}_{i=1}^{<k}$. We argue that predicting at a frame level and treating patch vectors of one observation as a whole better captures global structure and temporal dynamics, modeling dependencies across the entire observation rather than isolated tokens, leading to improved temporal generalization. The effectiveness of this attention mask has been shown in our ablation experiments in Appendix A.4.1

To model the effect of the agent's action to the environment, we condition the world model's predictions on these

actions. Specifically, we concatenate the $K$-dimensional action vector, mapped from the original action representation using a multi-layer perceptron (MLP), to each patch vector $z_t^i$ for $i = 1, \ldots, N$. When proprioceptive information is available, we incorporate it similarly by concatenating it to the observation latents, thereby integrating it into the latent states.

We train the world model with teacher forcing. During training, we slice the trajectories into segments of length $H + 1$, and compute a latent consistency loss on each of the $H$ predicted frames. For each frame, we compute

$$\mathcal{L}_{pred} = \|p_\theta\left(\text{enc}_\theta(o_{t-H:t}), \phi(a_{t-H:t})\right) - \text{enc}_\theta\left(o_{t+1}\right)\|^2 \tag{1}$$

where $\phi$ is the action encoder model that can map actions to higher dimensions. Note that our world model training is entirely performed in latent space, without the need to reconstruct the original pixel images.

### 3.1.3. DECODER FOR INTERPRETABILITY

To aid in visualization and interpretability, we use a stack of transposed convolution layers to decode the patch representations back to image pixels, similar as in (Razavi et al., 2019). Given a pre-collected dataset, we optimize the parameters $\theta$ of the decoder $q_\theta$ with a simple reconstruction loss defined as:

$$\mathcal{L}_{rec} = \|q_\theta(z_t) - o_t\|^2, \quad \text{where} \quad z_t = \text{enc}_\theta(o_t) \tag{2}$$

The training of the decoder is entirely independent of the transition model training, offering several advantages: **1)** The decoder does not affect the world model's reasoning and planning capabilities for solving downstream tasks, and **2)** There is no need to reconstruct raw pixel images during planning, thereby reducing computational costs. Nevertheless, the decoder remains valuable as it enhances the interpretability of the world model's predictions. While backpropagating this decoder loss to the predictor is possible, we ablate this choice and find that it negatively impacts performance compared to omitting the decoder loss. Full details are provided in Appendix A.4.2.

### 3.2. Visual Planning with DINO-WM

To evaluate the quality of the world model, we perform trajectory optimization at test time and measure performance. While the planning methods themselves are fairly standard, they serve as means to emphasize the quality of the world models. For this purpose, our world model receives the current observation $o_0$ and a goal observation $o_g$, both represented as RGB images. We formulate planning as the process of searching for a sequence of actions that the agent would take to reach $o_g$. We employ model predictive control (MPC), which facilitates planning by considering the outcomes of future actions.

We utilize the cross-entropy method (CEM) to optimize the sequence of actions at each iteration. The planning cost is defined as the mean squared error (MSE) between the current latent state and the goal's latent state, given by

$$\mathcal{C} = \|\hat{z}_T - z_g\|^2, \quad \text{where} \quad \begin{aligned} \hat{z}_t &= p(\hat{z}_{t-1}, a_{t-1}), \\ \hat{z}_0 &= \text{enc}(o_0), \\ z_g &= \text{enc}(o_g). \end{aligned}$$

The MPC framework and CEM optimization procedure are detailed in Appendix A.5.1. Since our world model is differentiable, a possibly more efficient approach is to optimize this objective through gradient descent (GD), allowing the world model to directly guide the agent toward a specific goal. The details of GD are provided in Appendix A.5.2. However, we empirically observe that CEM outperforms GD in our experiments with full results in Appendix A.5.3. We hypothesize that incorporating regularizations during training and in the planning objectives could further improve performance, and leave this for future work.

## 4. Experiments

Our experiments are designed to address the following key questions: **1)** Can we effectively train DINO-WM using precollected offline datasets? **2)** Once trained, can DINO-WM be used for visual planning? **3)** To what extent does the quality of the world model depend on pre-trained visual representations? **4)** Does DINO-WM generalize to new configurations, such as variations in spatial layouts and object arrangements? **5)** How does DINO-WM's performance scale with offline dataset size? We train and evaluate DINO-WM across six environment suites (full description in Appendix A.1), comparing it to state-of-the-art world models that predict in either latent space or raw pixel space.

### 4.1. Environments and Tasks

We evaluate six environment suites with varying dynamics complexity, some of which are drawn from standard robotics benchmarks, such as D4RL (Fu et al., 2021) and DeepMind Control Suite (Tassa et al., 2018), as shown in Figure 3. These environments include maze navigation (**Maze**, **Wall**), fine-grained control for tabletop pushing (**PushT**) and robotic arm control (**Reach**), and deformable object manipulation with an XArm (**Rope**, **Granular**).

In all environments, the task is to reach a randomly sampled goal state specified by a target observation, starting from arbitrary initial states. For PushT, target configurations are sampled to ensure feasibility within 25 steps. For Granular, targets require gathering all particles into a square with randomized locations and sizes. Observations in all environments are RGB images of size (224, 224). A full description of the environments is provided in Appendix A.1.

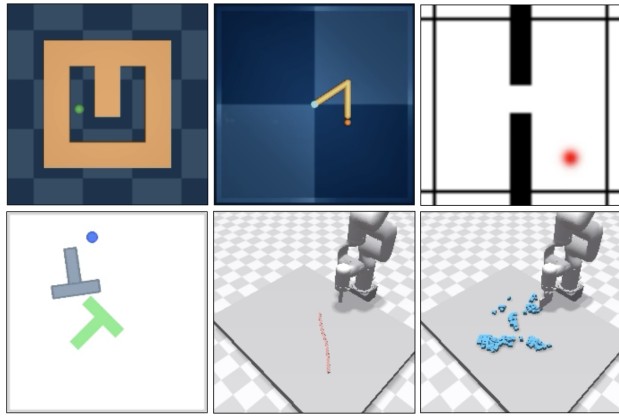

Figure 3. We evaluate DINO-WM on six environment suites, from left to right, top to bottom: Maze, Reach, Wall, Push-T, Rope Manipulation, and Granular Manipulation.

## 4.2. Baselines

We compare DINO-WM with the following state-of-the-art models commonly used for control. For IRIS, DreamerV3, and TD-MPC2, we train the models with our offline datasets without any reward or task information, and perform MPC on the learned world model for solving downstream tasks.

a) **IRIS (Micheli et al., 2023):** IRIS encodes visual inputs into tokens via a discrete autoencoder and predicts future tokens using a GPT Transformer, enabling policy and value learning through imagination.

b) **DreamerV3 (Hafner et al., 2024):** DreamerV3 encodes visual inputs into categorical representations, predicts future states and rewards, and trains an actor-critic policy from imagined trajectories.

c) **TD-MPC2 (Hansen et al., 2024) :** TD-MPC2 learns a decoder-free world model in latent space and uses reward signals to optimize the latents.

d) **AVDC (Ko et al., 2023):** AVDC uses a diffusion model to generate task execution videos from an initial observation and textual goal. We provide qualitative evaluations and MPC planning results for an action-conditioned variant in Section 4.6.

## 4.3. Optimizing Behaviors with DINO-WM

With a trained world model, we study if DINO-WM can be used for zero-shot planning directly in the latent space.

For Maze, Reach, PushT, and Wall environments, we sample 50 initial and goal states and measure the success rate across all instances. Due to the environment stepping time for the Rope and Granular environments, we evaluate the Chamfer

Distance (CD) on 10 instances for them. In Granular, we sample a random configuration from the validation set, with the goal of pushing the materials into a square shape at a randomly selected location and scale.

Table 1. Planning results for offline world models on six control environments.

| Model | Maze SR ↑ | Wall SR ↑ | Reach SR ↑ | PushT SR ↑ | Rope CD ↓ | Granular CD ↓ |
|---|---|---|---|---|---|---|
| IRIS | 0.74 | 0.04 | 0.18 | 0.32 | 1.11 | 0.37 |
| DreamerV3 | **1.00** | **1.00** | 0.64 | 0.30 | 2.49 | 1.05 |
| TD-MPC2 | 0.00 | 0.00 | 0.00 | 0.00 | 2.52 | 1.21 |
| Ours | 0.98 | 0.96 | **0.92** | **0.90** | **0.41** | **0.26** |

As seen in Table 1, on simpler environments such as Wall and PointMaze, DINO-WM is on par with state-of-art world models like DreamerV3. However, DINO-WM significantly outperforms prior work at manipulation environments where rich contact information and object dynamics need to be accurately inferred for task completion. We notice that for TD-MPC2, the lack of reward signal makes it difficult to learn good latent representations, which subsequently results in poor performance. Visualizations of planning on all environments can be found in Appendix A.8.

Does DINO-WM learn better environment dynamics as more data become available? We conduct a set of ablation experiments in Section 4.8, showing that the planning performance scales positively with the amount of training data. We also present the full inference and planning times for DINO-WM in Appendix A.6, showing significant speedup over traditional simulation, particularly in the computationally intensive deformable environments.

## 4.4. Does pre-trained visual representations matter?

We use different pre-trained general-purpose encoders as the observation model of the world model, and evaluate their downstream planning performance. Specifically, we use the following encoders commonly used in robotics control and general perception: R3M (Nair et al., 2022), ImageNet pretrained ResNet-18 (Russakovsky et al., 2015; He et al., 2016) and DINO CLS (Caron et al., 2021). Detailed descriptions of these encoders are in Appendix A.3.

Table 2. Planning results for world models with various pre-trained encoders.

| Model | Maze SR ↑ | Wall SR ↑ | Reach SR ↑ | PushT SR ↑ | Rope CD ↓ | Granular CD ↓ |
|---|---|---|---|---|---|---|
| R3M | 0.94 | 0.34 | 0.40 | 0.42 | 1.13 | 0.95 |
| ResNet | **0.98** | 0.12 | 0.06 | 0.20 | 1.08 | 0.90 |
| DINO CLS | 0.96 | 0.58 | 0.60 | 0.44 | 0.84 | 0.79 |
| DINOPatch (Ours) | **0.98** | **0.96** | **0.92** | **0.90** | **0.41** | **0.26** |

We report the planning performance in Table 2. In the Point-

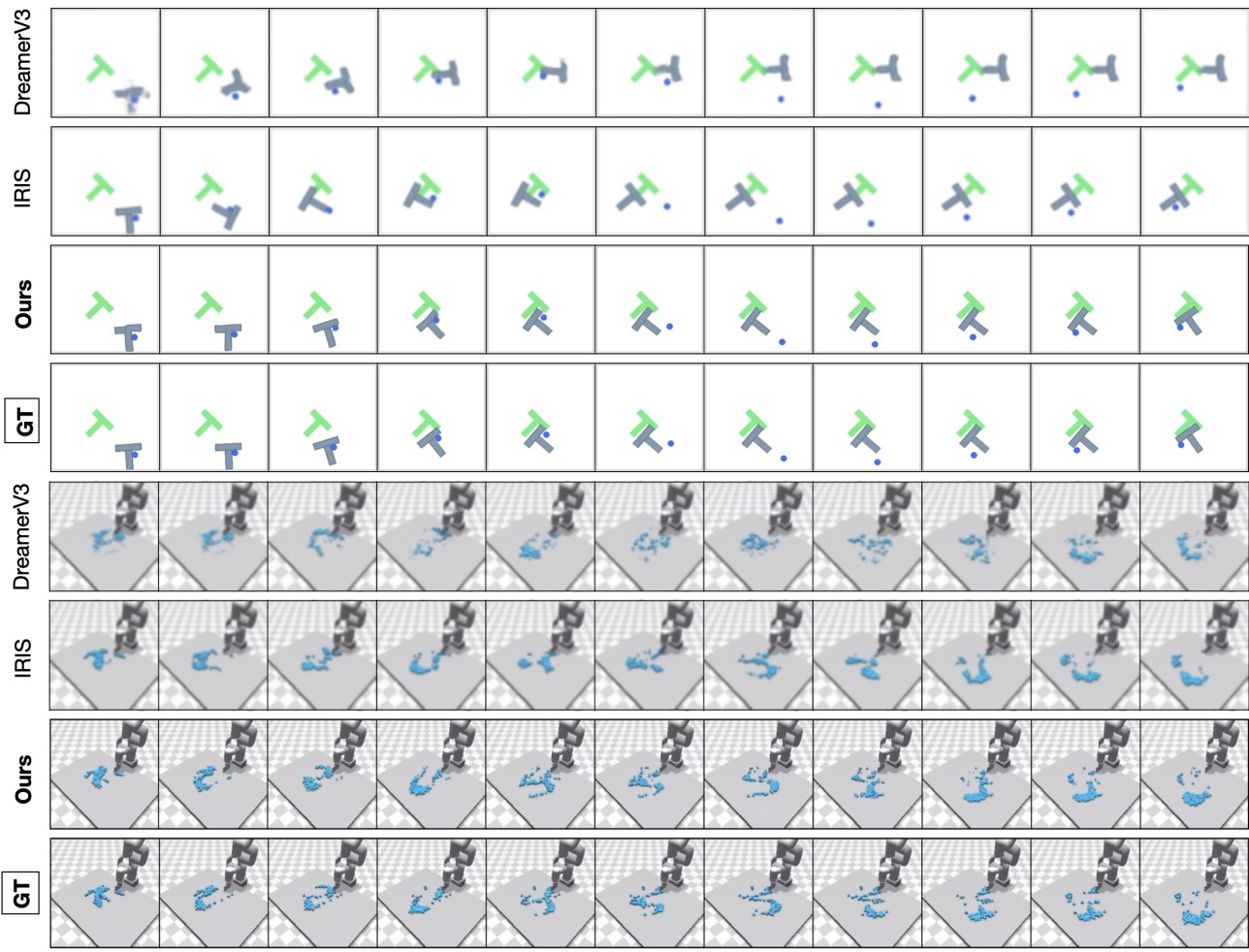

*Figure 4.* Open-loop rollouts of world models on Push-T and Granular. Given the first frame and action sequence, each model predicts future frames, reconstructed by its decoder. For each environment, the bottom row denotes the ground truth. DINO-WM (Ours) rollouts are bolded and are visually indistinguishable from the ground truth observations.

Maze task, which involves simple dynamics and control, we observe that world models with various observation encoders all achieve near-perfect success rates. However, as the environment's complexity increases—requiring more precise control and spatial understanding—world models that encode observations as a single latent vector show a significant drop in performance. We posit that patch-based representations better capture spatial information, in contrast to models like R3M, ResNet, and DINO CLS, which reduce observations to a single global feature vector, losing crucial spatial details necessary for manipulation tasks.

To better understand the strong planning performance enabled by DINOv2 features, we analyze the features directly. A well-established method for evaluating feature quality in downstream control tasks is linear probing from the features to environment states, which assesses how well the features encode task-relevant state information. To this end,

we compare the linear probe results on three environments across both patch-based features (DINOv2 of various ViT sizes, pre-trained MAE (He et al., 2021) and global features (DINO CLS, R3M). The validation loss for these linear probes is reported in Table 3, where DINO-S Patch and DINO-B Patch achieve the lowest validation loss, indicating their superior task representation capabilities. While the pre-trained MAE also has patch-based features, it has much higher validation loss. We hypothesize this is because MAE prioritizes reconstruction over task relevance, making it a less preferable choice for control tasks.

### 4.5. Generalizing to Novel Environment Configurations

We evaluate the generalization of our world models not only across different goals but also across various environment configurations. We construct three environment families—WallRandom, PushObj, and GranularRan-

*Table 3.* Linear Probe Validation Loss for Pre-trained Encoders. We evaluate the linear probe performance by mapping the embeddings from each encoder to the state vector of each environment. DINO-S and DINO-B denote DINOv2 models with ViT-Small and ViT-Base architectures, respectively. For patch-based features (DINO-S Patch, DINO-B Patch, and Pre-trained MAE), we first flatten the patch embeddings, project them to a 1536-dimensional vector, and then feed them into a linear probe model. Our results show that DINO-S Patch and DINO-B Patch achieve the lowest validation loss.

| Method | PointMaze | PushT | Wall |
|---|---|---|---|
| DINO-S Patch | 0.017 | **0.434** | 0.184 |
| DINO-B Patch | **0.014** | 0.504 | **0.163** |
| DINO-S CLS | 0.475 | 0.833 | 0.519 |
| Pre-trained MAE | 0.856 | 0.804 | 0.711 |
| R3M | 0.192 | 0.902 | 0.539 |

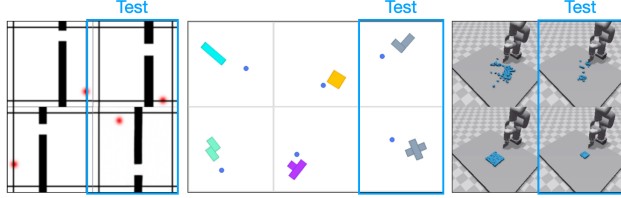

*Figure 5.* Training and testing setups for WallRandom, PushObj and GranularRandom. Test setups are highlighted in blue.

dom—where the model is tested on unseen configurations with random goals. Visualizations of training and testing examples are shown in Figure 5, and detailed descriptions of the environments can be found in Appendix A.2.

*Table 4.* Planning results for offline world models on three suites with unseen environment configurations.

| Model | WallRandom | PushObj | GranularRandom |
|---|---|---|---|
| | SR ↑ | SR ↑ | CD ↓ |
| IRIS | 0.06 | 0.14 | 0.86 |
| DreamerV3 | 0.76 | 0.18 | 1.53 |
| R3M | 0.40 | 0.16 | 1.12 |
| ResNet | 0.40 | 0.14 | 0.98 |
| DINO CLS | 0.64 | 0.18 | 1.36 |
| Ours | **0.82** | **0.34** | **0.63** |

From Table 4, we observe that DINO-WM demonstrates significantly better performance in WallRandom, indicating that model has effectively learned the general concepts of walls and doors, even when they are positioned in locations unseen during training. In contrast, other methods struggle to accurately identify the door's position and navigate through it. The PushObj task remains challenging for all methods, as the model was only trained on the four object shapes, which makes it difficult to precisely infer relevant physical parameters. In GranularRandom, the agent

encounters fewer than half the particles present during training, resulting in out-of-distribution images compared to the training instances. Nevertheless, DINO-WM accurately encodes the scene and successfully gathers the particles into a designated square location with the lowest Chamfer Distance (CD) compared to the baselines, demonstrating better scene understanding. We hypothesize that this is due to DINO-WM's observation model encoding the scene as patch features, making the variance in particle number still within the distribution for each image patch.

### 4.6. Qualitative Comparisons with Generative Video Models

Given the prominence of generative video models, it's natural to assume they could serve as world models. We compare DINO-WM with AVDC (Ko et al., 2023), a diffusion-based generative model. As shown in Figure 6, while AVDC can generate visually realistic future images, these images lack physical plausibility. Large, unrealistic changes can occur within a single timestep, and the model struggles to reach the exact goal state. Future advancements in generative models may help address these issues.

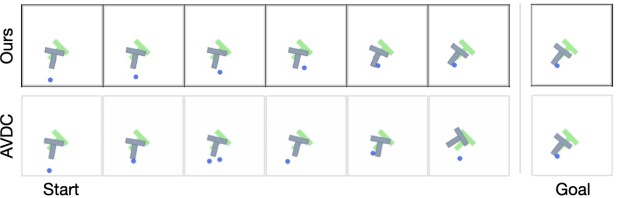

*Figure 6.* Plans generated by DINO-WM and AVDC.

We further compare DINO-WM with a variant of AVDC, where the diffusion model is trained to generate the next observation $o_{t+1}$ conditioned on the current observation $o_t$ and action $a_t$, rather than generating an entire sequence of observations at once conditioned on a text goal. We present open-loop rollout results on validation trajectories using this action-conditioned AVDC, with visualizations shown in Figure 7. It can be seen that the action-conditioned AVDC diverges from the ground truth observations over long-term predictions, making it insufficient for accurate task planning.

### 4.7. Decoding and Interpreting the Latents

Although DINO-WM operates in latent space and the observation model is not trained with pixel reconstruction objectives, training a decoder aids in interpreting predictions. We evaluate the image quality of predicted futures across all models and find that our approach outperforms others, even those whose encoders are trained with environment-specific reconstruction objectives. Open-loop rollouts in Figure 4 demonstrate DINO-WM's robustness despite the lack of explicit pixel supervision. We report the Learned Perceptual

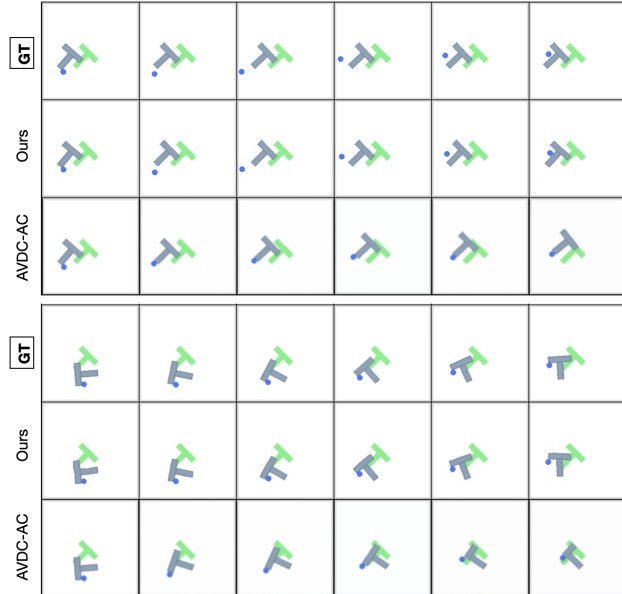

*Figure 7.* Open-loop rollout on PushT with DINO-WM and action-conditioned AVDC (AVDC-AC). For each trajectory, the model is given the first frame as well as sequence of actions. The world models perform open-loop rollout with these actions.

Image Patch Similarity (LPIPS) (Zhang et al., 2018) and Structural Similarity Index (SSIM) (Wang et al., 2004) on the world models' predicted future frames in Table 5 and Table 6. SSIM measures the perceived quality of images by evaluating structural information and luminance consistency between predicted and ground-truth images, with higher values indicating greater similarity. LPIPS assesses perceptual similarity by comparing deep representations of images, with lower scores reflecting closer visual similarity.

*Table 5.* Comparison of world model precitions on LPIPS (↓).

| Method | PushT | Wall | Rope | Granular |
|---|---|---|---|---|
| R3M | 0.045 | 0.008 | 0.023 | 0.080 |
| ResNet | 0.063 | 0.002 | 0.025 | 0.080 |
| DINO CLS | 0.039 | 0.004 | 0.029 | 0.086 |
| AVDC | 0.046 | 0.030 | 0.060 | 0.106 |
| Ours | **0.007** | **0.0016** | **0.009** | **0.035** |

*Table 6.* Comparison of world model precitions on SSIM (↑).

| Method | PushT | Wall | Rope | Granular |
|---|---|---|---|---|
| R3M | 0.956 | 0.994 | 0.982 | 0.917 |
| ResNet | 0.950 | 0.996 | 0.980 | 0.915 |
| DINO CLS | 0.973 | 0.996 | 0.980 | 0.912 |
| AVDC | 0.959 | 0.983 | 0.979 | 0.909 |
| Ours | **0.985** | **0.997** | **0.985** | **0.940** |

### 4.8. Scaling Laws of DINO-WM

To analyze the scaling behavior of DINO-WM, we trained world models and performed planning using datasets of varying sizes, ranging from 200 to 18500 trajectories on the PushT environment. Our results in Table 7 demonstrate a clear trend: as the dataset size increases, both the quality of the world model's predictions and the performance of the planned behavior improve significantly. Larger datasets enable the world model to capture more diverse dynamics and nuances of the environment, leading to more accurate predictions and better-informed planning.

*Table 7.* Planning performance and prediction quality on PushT with DINO-WM trained on datasets of various sizes. SSIM and LPIPS are measured on the predicted future latents after decoding. We observe consistent improvement in performance as we increase the dataset size.

| Dataset Size | SR ↑ | SSIM ↑ | LPIPS ↓ |
|---|---|---|---|
| n=200 | 0.08 | 0.949 | 0.056 |
| n=1000 | 0.48 | 0.973 | 0.013 |
| n=5000 | 0.72 | 0.981 | 0.007 |
| n=10000 | 0.88 | 0.984 | 0.006 |
| n=18500 | 0.92 | 0.987 | 0.005 |

## 5. Conclusion

We introduce DINO-WM, a simple yet effective technique for modeling visual dynamics in latent space without the need for pixel-space reconstruction. We have demonstrated that DINO-WM captures environmental dynamics and generalizes to unseen configurations, independent of task specifications, enabling visual reasoning at test time and generating zero-shot solutions for downstream tasks through planning. DINO-WM takes a step toward bridging the gap between task-agnostic world modeling and reasoning and control, offering promising prospects for generic world models in real-world applications.

**Limitations and Future Work**: First, DINO-WM assumes access to offline datasets with sufficient state-action coverage, which can be challenging to obtain for highly complex environments. This can potentially be addressed by combining DINO-WM with exploration strategies and updating the model as new experiences are available. Second, DINO-WM still relies on the availability of ground truth actions from agents, which may not always be feasible when training with vast video data from the internet. Lastly, while we currently plan in action space for downstream task solving, an extension of this work could involve developing a hierarchical structure that integrates high-level planning with low-level control policies to enable solving more fine-grained control tasks.

## Acknowledgements

We would like to thank Ademi Adeniji, Alfredo Canziani, Amir Bar, Kevin Zhang, Mido Assran, Vlad Sobal, Zichen Jeff Cui for their valuable discussion and feedback. This work was supported by grants from Honda, Hyundai, NSF award 2339096 and ONR awards N00014-21-1-2758 and N00014-22-1-2773. LP is supported by the Packard Fellowship.

## Impact Statement

This paper presents work whose goal is to facilitate the learning and applications of task-agnostic world models. There are many potential societal consequences of our work, none which we feel must be specifically highlighted here.

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

# A. Appendix

## A.1. Environments and Dataset Generation

a) **PointMaze:** In this environment introduced by (Fu et al., 2021), the task is for a force-actuated 2-DoF ball in the Cartesian directions $x$ and $y$ to reach a target goal. The agent's dynamics incorporate physical properties such as velocity, acceleration, and inertia, making the movement realistic. We generate 2000 fully random trajectories to train our world models. We refer to this task as **Maze** for brevity in our tables.

b) **Wall:** This custom 2D navigation environment features two rooms separated by a wall with a door. The agent's task is to navigate from a randomized starting location in one room to a goal in the other, passing through the door. We present a variant where wall and door positions are randomized, testing the model's generalization to novel configurations. For the fixed wall setting, we train on a fully random dataset of 1920 trajectories each with 50 time steps. For the variant with multiple training environment configurations, we generate 10240 random trajectories.

c) **Reacher:** A continuous control task from DeepMind Control Suite (Tassa et al., 2018), where a 2-joint robotic arm reaches a target in 2D space. We increase difficulty by requiring the entire arm, not just the end-effector, to match arbitrary target poses. To train the world model, we generate 3000 trajectories with 100 steps. We refer to this task as **Reach** for brevity in our tables.

d) **Push-T:** This environment introduced by (Chi et al., 2024) features a pusher agent interacting with a T-shaped block. The goal is to guide both the agent and the T-block from a randomly initialized state to a known feasible target configuration within 25 steps. The task requires both the agent and the T to match the target locations. Unlike previous setups, the fixed green T no longer represents the target position for the T-block but serves purely as a visual anchor for reference. Success requires precise understanding of the contact-rich dynamics between the agent and the object, making it a challenging test for visuomotor control and object manipulation. We generate a dataset of 18500 samples replayed the original released expert trajectories with various level of noise. Additionally, we introduce variations by altering the shape and color of the object to assess the model's capability to adapt to novel tasks. For this variant, we generate 20000 randomly sampled trajectories with 100 steps.

e) **Rope Manipulation:** Introduced in (Zhang et al., 2024), this task is simulated with Nvidia Flex (Zhang et al., 2024) and consists of an XArm interacting with a soft rope placed on a tabletop. The objective is to move the rope from an arbitrary starting configuration to a goal configuration specified at test time. For training, we generate a random dataset of 1000 trajectories of 20 time steps of random actions from random starting positions, while testing involves goal configurations set from varied initial positions, incorporating random variations in orientation and spatial displacement.

f) **Granular Manipulation:** This environment uses the same simulation setup as Rope Manipulation and involves manipulating about a hundred particles to form desired shapes. The training data consists of 1000 trajectories of 20 time steps of random actions starting from the same initial configuration, while testing is performed on specific goal shapes from diverse starting positions, along with random variations in particle distribution, spacing, and orientation.

## A.2. Environment Families for Testing Generalization

1. **WallRandom:** Based on the Wall environment, but with randomized wall and door positions. At test time, the task requires navigating from a random starting position on one side of the wall to a random position on the other side, with non-overlapping wall and door positions seen during training.

2. **PushObj:** Derived from the Push-T environment, where we introduce novel block shapes, including Tetris-like blocks and a "+" shape. We train the model with four shapes and evaluate on two unseen shapes. The task involves both the agent and object reaching target locations.

3. **GranularRandom:** Derived from the Granular environment, where we initialize the scene with a different amount of particles. The task requires the robot to gather all particles to a square shape at a randomly sampled location. For this task, we directly take the models that are trained with a fixed amount of materials used in Section 4.3.

Visualizations can be found in Figure 5.

## A.3. Pretraining Features

a) **R3M:** A ResNet-18 model pre-trained on a wide range of real-world human manipulation videos (Nair et al., 2022).

b) **ImageNet:** A ResNet-18 model pre-trained on the ImageNet-1K dataset (Russakovsky et al., 2015).

c) **DINO CLS:** The pre-trained DINOv2 model provides two types of embeddings: Patch and CLS. The CLS embedding is a 1-dimensional vector that encapsulates the global information of an image.

d) **Pre-trained MAE:** A ViT model trained using Masked Autoencoding (He et al., 2021), where a large portion of input image patches are masked and the model learns to reconstruct them. We use a ViT-Base checkpoint, which has a feature dimension of 768 and approximately 86 million parameters.

## A.4. Ablations

### A.4.1. DINO-WM WITH VS. WITHOUT CAUSAL ATTENTION MASK

We introduce a causal attention mask in Section 3.1.2. We ablate this choice on PushT by training DINO-WM with and without this causal attention mask with varying history length $h$, such that the model takes in input $o_{t-h+1}, o_{t-h+2}, ...o_t$, and output $o_{t-h+2}, ...o_{t+1}$. For models *with mask*, the model can only attend to past observations for predicting each $o_t$, whereas in the *w/o mask* case, predicting any observation in the output sequence can attend to the entire input sequence of observations. We show planning success rate on our PushT settings in Table 8. When $h = 1$ where the model with and without this causal mask is equivalent, both models get decent and equivalent success rate. However, as we increase the history length, we see a rapid drop in the *w/o mask* case, since the model can cheat during training by attending to future frames, which is not available at test time. Adding the causal mask solves this issue, and we observe improvement in performance as longer history could better capture dynamics information like velocity, acceleration, and object momentum.

*Table 8.* Comparison of DINO-WM with and without causal attention mask on PushT. We train models with varying history $h$, representing the number of past observations the model takes as input.

|           | $h = 1$ | $h = 2$ | $h = 3$ |
|-----------|---------|---------|---------|
| w/o mask  | 0.76    | 0.36    | 0.08    |
| with mask | 0.76    | 0.88    | 0.92    |

### A.4.2. DINO-WM WITH RECONSTRUCTION LOSS

While DINO-WM eliminates the need to train world models with a pixel reconstruction loss—avoiding the risk of learning features irrelevant to downstream tasks—we conduct an ablation study where the predictor is trained using a reconstruction loss propagated from the decoder. As shown in Table 9, this approach performs reasonably well on the PushT task but falls slightly short of our method, where the predictor is trained entirely independently of the decoder. This underscores the advantage of disentangling feature learning from reconstruction objectives.

*Table 9.* Comparison of DINO-WM trained with and without loss from the decoder on PushT, highlighting the advantage of disentangling feature learning from reconstruction objectives.

|                   | Success Rate |
|-------------------|--------------|
| w/o decoder loss  | 0.92         |
| with decoder loss | 0.80         |

## A.5. Planning Optimization

In this section, we detail the optimization procedures for planning in our experiments.

A.5.1. MODEL PREDICTIVE CONTROL WITH CROSS-ENTROPY METHOD

a) Given the current observation $o_0$ and the goal observation $o_g$, both represented as RGB images, the observations are first encoded into latent states:

$$\hat{z}_0 = \text{enc}(o_0), \quad z_g = \text{enc}(o_g). \tag{3}$$

b) The planning objective is defined as the mean squared error (MSE) between the predicted latent state at the final timestep $T$ and the goal latent state:

$$\mathcal{C} = \|\hat{z}_T - z_g\|^2, \quad \text{where} \quad \hat{z}_t = p(\hat{z}_{t-1}, a_{t-1}), \quad \hat{z}_0 = \text{enc}(o_0). \tag{4}$$

c) At each planning iteration, CEM samples a population of $N$ action sequences, each of length $T$, from a distribution. The initial distribution is set to be Gaussian.

d) For each sampled action sequence $\{a_0, a_1, \ldots, a_{T-1}\}$, the world model is used to predict the resulting trajectory in the latent space:

$$\hat{z}_t = p(\hat{z}_{t-1}, a_{t-1}), \quad t = 1, \ldots, T. \tag{5}$$

And the cost $\mathcal{C}$ is calculated for each trajectory.

e) The top $K$ action sequences with the lowest cost are selected, and the mean and covariance of the distribution are updated accordingly.

f) A new set of $N$ action sequences is sampled from the updated distribution, and the process repeats until success is achieved or after a fixed number of iterations that we set as hyperparameter.

g) After the optimization process is done, the first $k$ actions $a_0, ...a_k$ is executed in the environment. The process then repeats at the next time step with the new observation.

A.5.2. GRADIENT DESCENT:

Since our world model is differentiable, we also consider an optimization approach using Gradient Descent (GD) which directly minimizes the cost by optimizing the actions through backpropagation.

a) We first encode the current observation $o_0$ and goal observation $o_g$ into latent spaces:

$$\hat{z}_0 = \text{enc}(o_0), \quad z_g = \text{enc}(o_g). \tag{6}$$

b) The objective remains the same as for CEM:

$$\mathcal{C} = \|\hat{z}_T - z_g\|^2, \quad \text{where} \quad \hat{z}_t = p(\hat{z}_{t-1}, a_{t-1}), \quad \hat{z}_0 = \text{enc}(o_0). \tag{7}$$

c) Using the gradients of the cost with respect to the action sequence $\{a_0, a_1, \ldots, a_{T-1}\}$, the actions are updated iteratively:

$$a_t \leftarrow a_t - \eta \frac{\partial \mathcal{C}}{\partial a_t}, \quad t = 0, \ldots, T-1, \tag{8}$$

where $\eta$ is the learning rate

d) The process repeats until a fixed number of iterations is reached, and we execute the first $k$ actions $a_0, ..., a_k$ in the enviornment, where $k$ is a pre-determined hyperparameter.

A.5.3. PLANNING RESULTS

Here we present the full planning performance using various planning optimization methods in Table 10. CEM denotes the setting where we use CEM to optimize a sequence of actions, and execute those actions in the environment without any correction or replan. Similarly, GD denotes optimizing with gradient decent and execute all planned actions at once in an open-loop way. MPC denotes allowing replan and receding horizon with CEM for optimization.

Table 10. Planning results of DINO-WM with various planning optimization methods.

| | PointMaze | Push-T | Wall | Rope | Granular |
|---|---|---|---|---|---|
| CEM | 0.8 | 0.86 | 0.74 | NA | NA |
| GD | 0.22 | 0.28 | NA | NA | NA |
| MPC | 0.98 | 0.90 | 0.96 | 0.41 | 0.26 |

## A.6. Inference Time

Inference time is a critical factor when deploying a model for real-time decision-making. Table 11 reports the time required on an NVIDIA A6000 GPU for a single inference step, the environment rollout time for advancing one step in the simulator, and the overall planning time for generating an optimal action sequence using the Cross-Entropy Method (CEM). The inference time of DINO-WM remains constant across environments due to the fixed model size and input image resolution, resulting in significant speedup over traditional simulation rollouts. Notably, in environments with high computational demands, such as deformable object manipulation, simulation rollouts require several seconds per step while DINO-WM enables rapid inference and efficient planning. Planning time is measured with CEM using 100 samples per iteration and 10 optimization steps, demonstrating that DINO-WM can achieve feasible planning times while maintaining accuracy and adaptability across tasks.

Table 11. Inference time and planning time for DINO-WM. Inference time represents the time required for a single forward pass for one step, while environment rollout time measures the simulator's speed for advancing one step. Planning time corresponds to Cross-Entropy Method (CEM) with 100 samples per iteration and 10 optimization steps.

| Metric | Time (s) |
|---|---|
| Inference (Batch 32) | 0.014 |
| Simulation Rollout (Batch 1) | 3.0 |
| Planning (CEM, 100x10) | 15.89 |

## A.7. Hyperparameters and Implementation

We present the DINO-WM hyperparameters and relevant implementation repos below. We train the world models for all environments with the same hyperparameters shown in Table 13.

The world model architecture is consistent across all environments. We use an encoder based on DINOv2, which extracts features with a shape of $(14 \times 14, 384)$ from input images resized to $196 \times 196$ pixels. The ViT backbone has a depth of 6, 16 attention heads, and an MLP dimension of 2048, amounting to approximately 19M parameters.

To ensure the prediction task is meaningful, as nearby observations can be highly similar, we introduce a frameskip parameter during data processing. This parameter specifies how far into the future the model is predicting. The frameskip values for each environment are provided in Table 12.

- DINOv2: https://github.com/facebookresearch/dinov2
- DreamerV3: https://github.com/NM512/dreamerv3-torch
- AVDC: https://github.com/flow-diffusion/AVDC
- R3M: https://github.com/facebookresearch/r3m/

We base our predictor implementation on https://github.com/lucidrains/vit-pytorch/.

## A.8. Additional Planning Visualizations

We show visualizations of planning instances for DINO-WM and our baselines in Figure 8. For comparison, we show the best performing world models DINO CLS and DreamerV3. We also show visualizations of DINO-WM on all tasks in

*Table 12.* Environment-dependent hyperparameters for DINO-WM training. We report the number of trajectories in the dataset under *Dataset Size*, and the length of trajectories under *Traj. Len.*

| | $H$ | Frameskip | Dataset Size | Traj. Len. |
|---|---|---|---|---|
| PointMaze | 3 | 5 | 2000 | 100 |
| Reacher | 3 | 5 | 3000 | 100 |
| Push-T | 3 | 5 | 18500 | 100-300 |
| PushObj | 3 | 5 | 20000 | 100 |
| Wall | 1 | 5 | 1920 | 50 |
| WallRandom | 1 | 5 | 10240 | 50 |
| Rope | 1 | 1 | 1000 | 5 |
| Granular | 1 | 1 | 1000 | 5 |

*Table 13.* Shared hyperparameters for DINO-WM training

| Name | Value |
|---|---|
| Image size | 224 |
| Optimizer | AdamW |
| Decoder lr | 3e-4 |
| Predictor lr | 5e-5 |
| Action encoder lr | 5e-4 |
| Action emb dim | 10 |
| Epochs | 100 |
| Batch size | 32 |

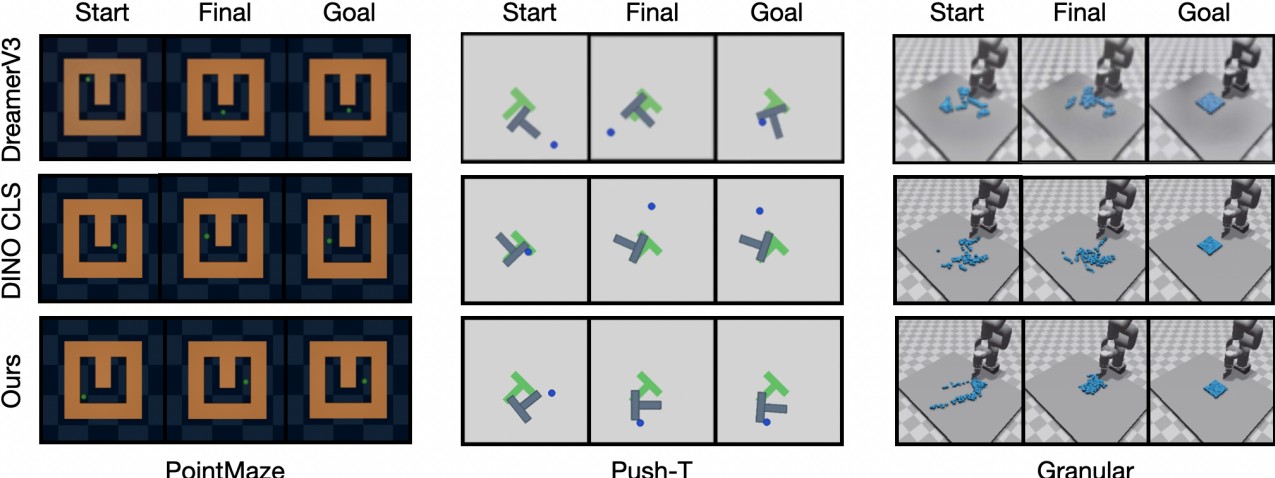

*Figure 8.* Planning visualizations for PointMaze, Push-T, and Granular, on randomly sampled initial and goal configurations. The task is defined by Start and Goal, denoting the initial and goal observations. Final shows the final state the system arrives at after planning with each world model. For comparison, we show the best performing world models DINO CLS and DreamerV3.

Figure 9. For each environment, the top (shaded) row shows the environment's observation after executing the planned actions, and the bottom row shows the world model's imagined observations.

To demonstrate DINO-WM's ability to generalize to different goals at test time, we show additional visualizations for DINO-WM when provided with the same initial observation but different goal observations in Figure 10 and Figure 11. Similarly, we show trajectory pairs to compare the environment's observations (top shaded rows) after executing a sequence of planned actions with DINO-WM's imagined trajectories (bottom rows). The left-most column denotes the initial observations, and the right-most shaded column denotes the goal observations.

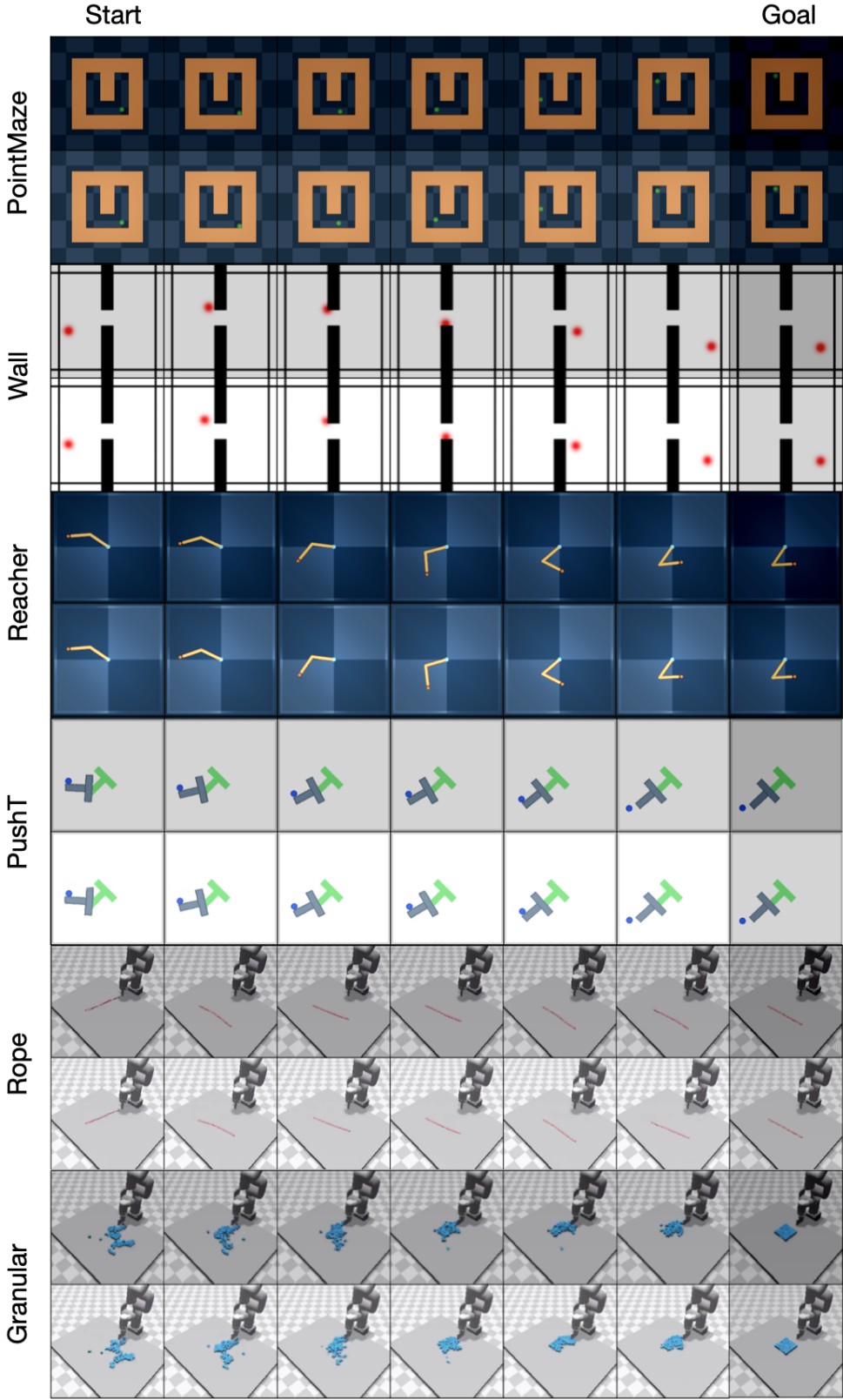

Figure 9. Trajectories planned with DINO-WM on all six environments. For each environment, the top (shaded) row shows the environment's observation after executing the planned actions, and the bottom row shows the world model's imagined observations.

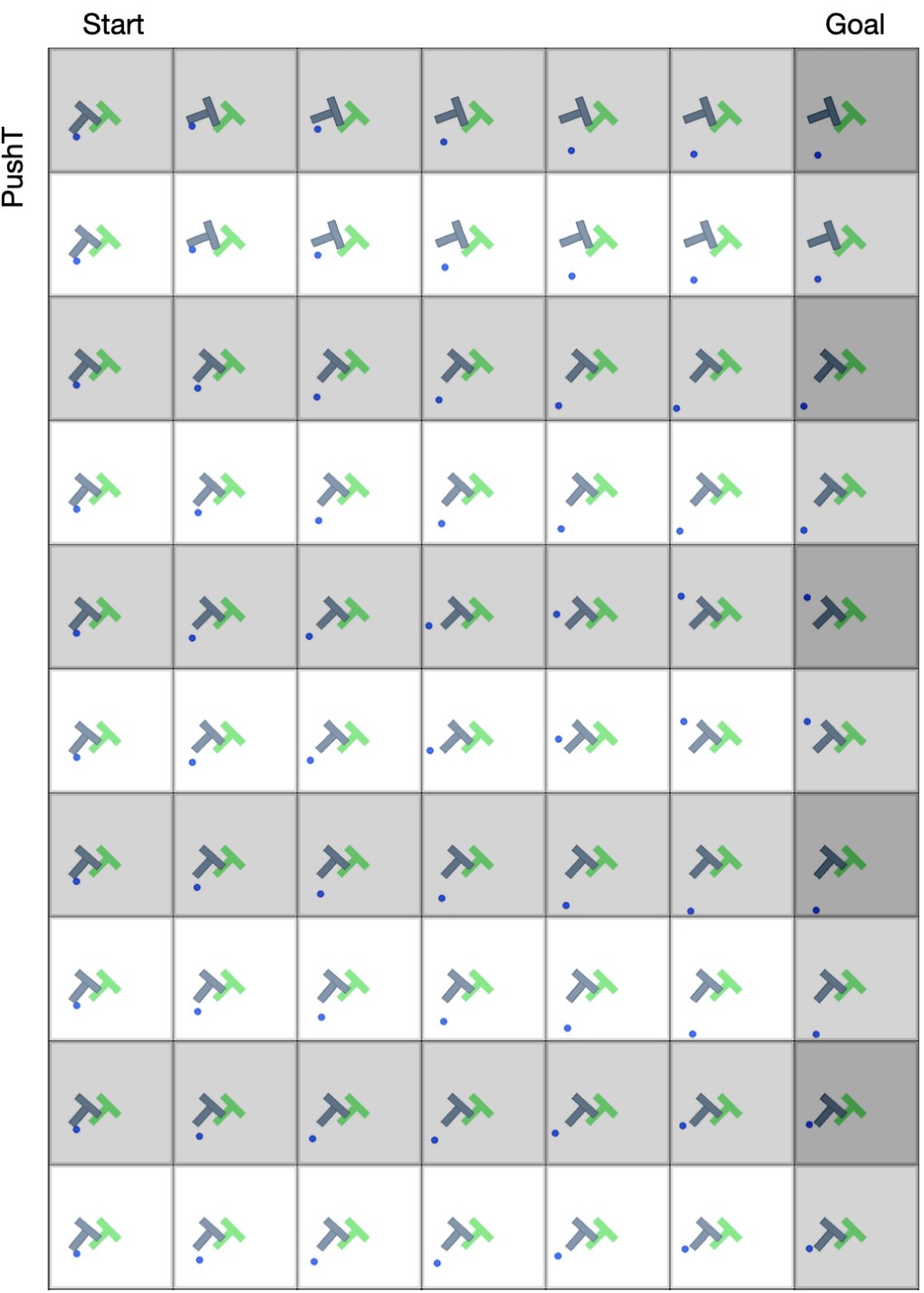

*Figure 10.* Trajectories planned with DINO-WM on PushT with the same initial states but different goal states.

Start                                                                                                                    Goal

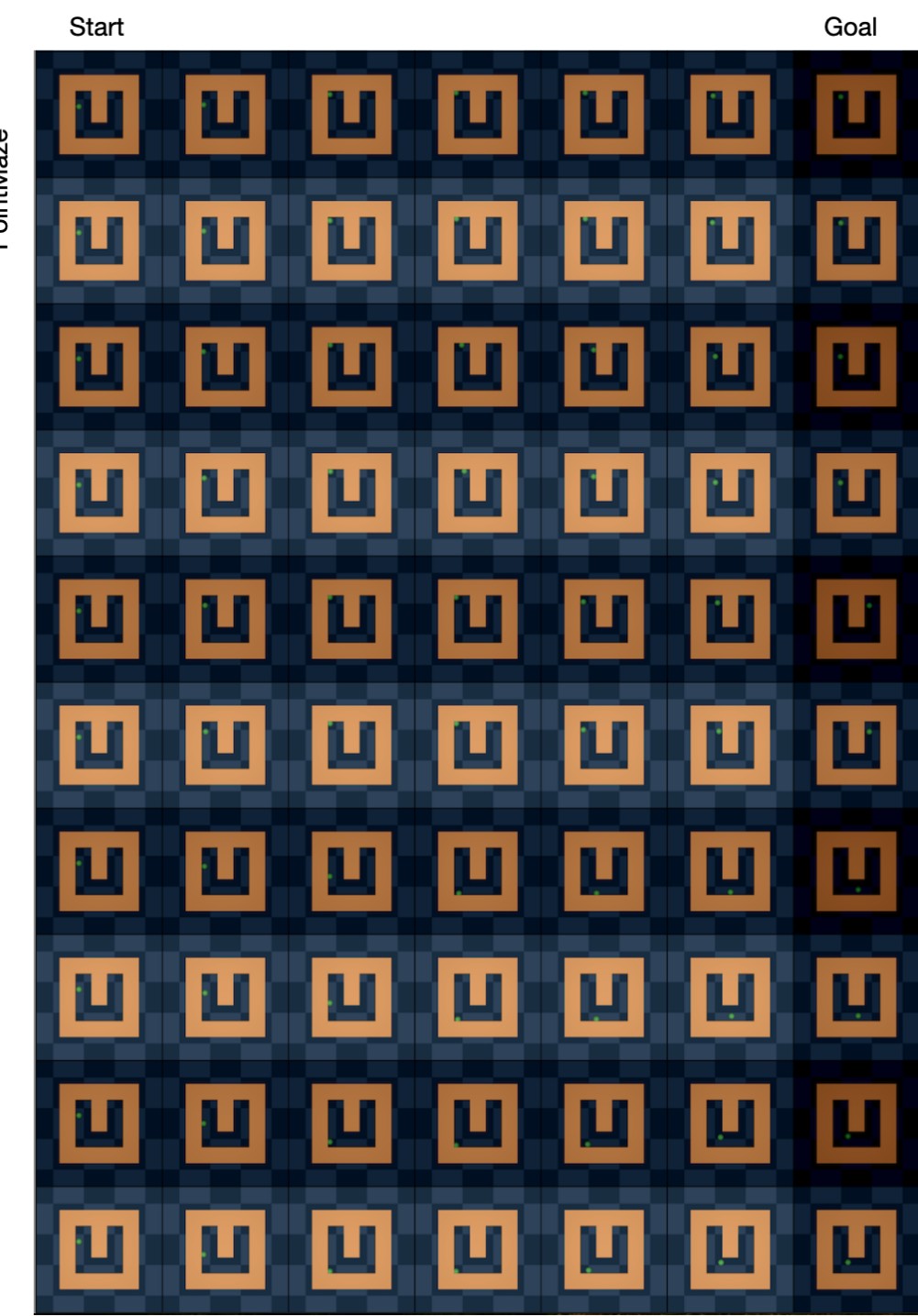

*Figure 11.* Trajectories planned with DINO-WM on PointMaze with the same initial states but different goal states.

