# OpenReview forum: "DINO-WM: World Models on Pre-trained Visual Features enable Zero-shot Planning"
_ICML.cc/2025/Conference — ICML 2025 poster_

### Official Review · Reviewer_Bqcm · 2025-03-12

**Overall Recommendation:** 3

**Summary:**

The paper introduces DINO‐WM, building task‐agnostic world models for control and planning. Instead of operating directly in pixel space, the method leverages pre‐trained patch features from DINOv2 to encode observations into a rich, spatially-aware latent representation. The world model is trained offline using trajectories and employs a ViT-based transition model to predict future latent states. At test time, planning is carried out using model predictive control (MPC) with the cross-entropy method (CEM), allowing for zero-shot goal-reaching without additional reward signals or task-specific tuning.

**Claims And Evidence:**

Yes

**Essential References Not Discussed:**

No.

**Ethics Expertise Needed:**

["Discrimination / Bias / Fairness Concerns"]

**Experimental Designs Or Analyses:**

The experimental setups are a bit easy. More experiments on tasks with complex textured backgrounds need to be evaluated.

**Methods And Evaluation Criteria:**

This proposed principle holds when facing multiple experimental settings. However, I am a little worried about the real-world planning and control performance since the robot is always moving when the proposed model is planning the actions. If the planning is too slow (53s), the control system could collapse.

**Other Comments Or Suggestions:**

No

**Other Strengths And Weaknesses:**

## Strengths :

The approach allows for planning at test time without task-specific fine-tuning or the need for expert demonstrations. The experiments show that DINO-WM can generate effective control policies purely through offline training, which is a significant step toward more general-purpose models.

The authors validate their method on six simulation environments (e.g., maze navigation, push manipulation, robotic arm control, and deformable object manipulation). The extensive comparisons with several state-of-the-art baselines (IRIS, DreamerV3, TD-MPC2, among others) highlight the method’s superior performance in both success rates and reconstruction quality.

The model demonstrates robust generalization, successfully handling novel environment configurations and object variations.

## Weakness
DINO-WM requires a comprehensive offline dataset with sufficient state-action coverage. In real-world settings, gathering such data might be challenging, potentially limiting the method’s applicability outside of controlled environments.

All experiments are conducted in simulated tasks. Although the results are promising, further validation on real-world robotic platforms would be needed to assess practical deployment, as mentioned before.

The current planning framework operates solely in the action space. While effective for the tasks presented, incorporating hierarchical planning or multi-level control strategies could enhance performance on more complex or fine-grained tasks.

Although the paper includes ablation studies, additional analysis on hyperparameter sensitivity and computational trade-offs would provide deeper insights into the method’s robustness and scalability.

(Appendix A.4.2 is total a waste. Failed to understand why the authors want to add this paragraph.)

**Questions For Authors:**

More foundation models could be evaluated, like OpenVLA using Dino+Siglip. It would add some value if authors could investigate such setups.

**Relation To Broader Scientific Literature:**

The model demonstrates robust generalization, successfully handling novel environment configurations and object variations. This suggests that the learned latent representations capture essential underlying dynamics beyond the specific scenarios seen during training.

**Theoretical Claims:**

Theoretical claims are good.

---

> ### Author Rebuttal · Authors · 2025-04-01
>
> We thank the reviewer for their insightful review, especially in highlighting that DINO-WM demonstrates “robust generalization” and “is a significant step toward more general-purpose models.” We now address the issues raised by the reviewer.
>
> >**“I am a little worried about the real-world planning and control performance since the robot is always moving when the proposed model is planning the actions”**
>
> Our planning framework follows the principles of Model Predictive Control (MPC)—a well-established approach in real-world robotics. The planner optimizes future actions but executes only a subset before re-planning. The robot’s continuous movement is not an issue, as each planning step integrates updated proprioceptive and visual observations, ensuring robust closed-loop control.
>
> For planning efficiency, we improved our inference code for DINO-WM since submission and now it takes 15.89s for CEM compared to 53s reported in the manuscript. We provide further ways to speed up planning and we refer to our response to **Reviewer 57WD** due to space limits.
>
> >**“More experiments on tasks with complex textured backgrounds need to be evaluated.”**
>
> We evaluated ClutteredPushT, which adds a complex textured background to PushT. Please see our response to **Reviewer 57WD** for open-loop rollouts and final performance due to space limits.
>
> >**“DINO-WM requires a comprehensive offline dataset with sufficient state-action coverage” “validation on real-world robotic platforms”**
>
> We agree with the reviewer that DINO-WM requires a diverse dataset for effective planning. While it avoids online interactions, reward signals, and expert demonstrations, it still follows the general trade-off that broader coverage improves generalization. However, this limitation can be mitigated by continuously updating the world model with new experiences, enabling progressive improvement without requiring extensive real-world interactions upfront.
>
> For real-world validation, independent researchers have already successfully deployed DINO-WM on real robots. We are consulting with the AC on how to share this while maintaining anonymity.
>
> >**“incorporating hierarchical planning or multi-level control strategies could enhance performance on more complex or fine-grained tasks.”**
>
> We totally agree that hierarchical planning can unlock further capabilities of DINO-WM. Our current work serves as a proof of concept that even single-level planning can be effective with pre-trained WMs. This establishes a strong foundation on which more advanced planning strategies can be built. We see this as an exciting and promising direction for future work.
>
> >**“hyperparameter sensitivity and computational trade-offs” “method’s robustness and scalability”**
>
> We conducted a hyperparameter sweep for CEM on PushT. We refer to our response to **Reviewer 57WD** for the computation time and performance tradeoff due to space limits.
>
> For WM training, we ablated image sizes on PointMaze (see response to **Reviewer 4KWu**). The training hyperparameters for DINO-WM is quite robust, as the same hyperparameters work across all six reported environments.
>
> >**“Appendix A.4.2 is total a waste.”**
>
> Appendix A.4.2 demonstrates the necessity of the frame-level attention mask in DINO-WM’s predictor. The experiments (L749-753) supports our claim that the causal mask effectively ensures attention to past frames only, enabling the model to capture essential temporal dynamics such as velocity and acceleration. Would the reviewer prefer that we remove this section?
>
> >**“More foundation models could be evaluated”**
>
> We agree that evaluating more foundation models can provide further insights on the pros and cons of DINO-WM compared to existing models. To this end, we evaluated Genie, a foundation model for generating interactive environments. We train the Genie model on our PushT dataset (open-loop rollouts in [Figure 5](https://tinyurl.com/3659ydkn)). It is evident that Genie performs worse in prediction quality and future state estimation compared to DINO-WM, even with ground truth action conditioning.
>
> | Model         | LPIPS  |
> |--------------|--------|
> | Genie  | 0.043  |
> | Ours         | 0.007  |
>
> For OpenVLA with SigLIP, we note that it is a language-conditioned feed-forward policy trained with expert trajectories, fundamentally differing from DINO-WM, which learns environment dynamics from any interaction dataset and performs goal-conditioned planning. Since our setting doesn’t assume expert data or language conditioning, OpenVLA is unsuitable for fine-tuning. Additionally, its released model assumes a fixed action space, making it infeasible to finetune the model on our datasets.
>
> We also evaluate other foundational vision models such as pre-trained MAE as the encoder for DINO-WM. Please refer to our response to **Reviewer 4KWu** for the experiment details and final performance of this experiment due to the 5000 char response limit.

---

> > ### Comment · Reviewer_Bqcm · 2025-04-03
> >
> > Thanks for the rebuttal. I may not express it clearly, but my original intention is not to compare with openvla. Instead, I hope the authors can explore the foundation feature combination more, such as DINO+SigLip.
> >
> > I could raise my score, but I really would love to see results based on fusing multiple foundation features.

---

> > > ### Author Response · Authors · 2025-04-05
> > >
> > > Thank you for clarifying your question. We conducted additional experiments comparing models with single features (DINOv2, SigLip) and combined features (DINOv2 + SigLIP) following your suggestions. Results on the PushT environment are shown in the table below:
> > > | Model            | Feature Dim | Predictor Size | CEM  | MPC  | SSIM  | LPIPS |
> > > |------------------|-------------|----------------|------|------|-------|-------|
> > > | DINOv2 (Ours)    | 384         | 20,195,320      | 0.86 | 0.90 | 0.985 | 0.007 |
> > > | SigLIP           | 768         | 39,237,352      | 0.56 | 0.78 | 0.985 | 0.009 |
> > > | DINOv2 + SigLIP  | 1152        | 58,352,104      | 0.60 | 0.84 | 0.980 | 0.017 |
> > >
> > >
> > > We hypothesize that incorporating SigLIP features alongside DINOv2 did not improve performance because our tasks do not benefit from the language grounding that SigLIP offers. In fact, the combined DINOv2 + SigLIP features slightly underperform the DINO-only baseline for the final task planning success rate (MPC). This may be due to the significantly larger embedding space, which requires a larger predictor and can be harder to train effectively with a fixed dataset size.
> > >
> > > We hope this addresses your question, and we are happy to discuss any further questions you may have.

---

### Official Review · Reviewer_dmt6 · 2025-03-13

**Overall Recommendation:** 4

**Summary:**

This work aims to train a world model using large vision pre-trained features on offline trajectories in a task-agnostic fashion. They use the resulting trained world model to plan out Push-T, Maze, Reach, Rope, and Granalur control tasks in a zero-shot manner. Specifically, the authors train a world model using DINOv2 pre-trained features in the latent space. With extensive experimentation, the authors clearly show the benefit of using such a world model based on leveraging pre-trained large vision model features as opposed to the norm in the field -- to train the encoder from scratch.

**Claims And Evidence:**

I find all the claims made in the paper to be clear and supported by concrete experimental evidence.

**Essential References Not Discussed:**

None.

**Experimental Designs Or Analyses:**

Yes, and I don't have any particular concern with any experimental designs/analyses.

**Methods And Evaluation Criteria:**

Yes, the authors evaluate both plannings using the learned world model on (a) tasks on whose data they were trained on, (b) on novel unseen environment configurations -- testing the generalization capabilities as well as reporting visual similarity metrics such as LPIPS and SSIM when trained on a separate (and optional) decoder.

**Other Comments Or Suggestions:**

1. I find the choice of TDMPC2 as a baseline to be slightly strange. I believe TDMPC2 is learning a trivial (zero) representation because the only source for the TDMPC2-like world model to learn something meaningful is the reward. In the absence of reward, the "consistency" loss should immediately learn all zeros. Only reconstruction-based works like IRIS and DreamerV3 make sense as a baseline for this work. However, I haven't penalized the authors for this at all -- this was just a curious thought when I encountered the baseline.

**Other Strengths And Weaknesses:**

**Strengths**: I find the idea of using a pre-trained representation such as DINOv2's patch embeddings to be a really neat and simple idea to enable zero-shot planning. Leveraging the strengths of the large pre-trained visual models is an appealing research direction in contrast to encoders in MBRL/world model literature that have been trained from scratch.

**Weaknesses**: The authors can consider showing DINO-WM results on more robotic tasks (something like MetaWorld suite). I understanding that CEM planning over say picking or placing an object might be slightly challenging -- but it would be nice to see how DINO-WM does on those tasks. Even if it ends up failing, a discussion section describing the failures and potential reasons of that failure would be really helpful for the broader scientific community.

**Questions For Authors:**

1. What is the architecture that is used for the (optional) decoder to generate the results in Figure 4? Is the decode restricted to be the same across all the baselines (IRIS and DreamerV3) -- in which case I'd like the authors to report the number of parameters each decoder has. It would be ideal to showcase the results with the same decoder structure or with similar parameters to ensure that it is indeed the representation that is being evaluated.

**Relation To Broader Scientific Literature:**

I believe this work expands the current MBRL setting by incorporating the learned visual features from large vision models. Specifically training a world model on a task-agnostic dataset and being able to zero-shot plan on control tasks is a fundamental ability that a world model should possess. Although this paper does not evaluate on more complicated robotic tasks -- I believe this is a step in the right direction.

**Theoretical Claims:**

No theory is involved in this paper.

---

> ### Author Rebuttal · Authors · 2025-04-01
>
> We thank the reviewer for their insightful and constructive feedback, and for acknowledging that DINO-WM “expands the current MBRL setting by incorporating the learned visual features from large vision models”, and the idea of using pretrained features to be “neat and simple”. We now address the questions raised in the review below.
>
>
> **DINO-WM results on more robotic tasks**
>
> In addition to the robotic deformable manipulation environments and the robotic arm reaching task presented in the manuscript, we further provide results of DINO-WM on LIBERO [1], a tabletop environment manipulating diverse objects, with third-person image observations and a 7DoF action space. We note that CEM in this raw action space would be inefficient starting from completely random action samples (which would result in the robot mostly jittering in place). Therefore, instead of sampling randomly, we sample from a pre-trained diffusion policy [2]. DINO-WM is capable of long-horizon predictions and assists in selecting the best action trajectories. We provide visualizations for open-loop rollouts of DINO-WM in [Figure 1](https://tinyurl.com/5eckpz75) and videos for task planning [here](https://tinyurl.com/yptacs64). Without planning with DINO-WM, the diffusion policy model achieves a 35% success rate (across 20 trajectories). With planning using DINO-WM, the task success rate is improved to 55%. This demonstrates the effectiveness of DINO-WM, even with a high-dimensional continuous action space and a long task horizon.
>
>
> **TD-MPC2 as a baseline**
>
> We completely agree with the reviewer’s insight on why TD-MPC2 underperforms compared to other reconstruction-based baselines, as we have also discussed in the original manuscript (L290-293). Our motivation for including TD-MPC2 as a baseline is that it represents state-of-the-art work in world modeling while also incorporating planning to sample actions during training. We believe that comparing against TD-MPC2 provides valuable insight into the extent to which the performance of world models in this line of work depends on informative reward functions.
>
> **Architecture and size for decoders**
>
> We thank the reviewer for raising this constructive question. For the (optional) DINO-WM decoder, the architecture is based on VQ-VAE and consists of two stacked Decoder modules. Each Decoder is a transposed CNN with residual blocks. For DreamerV3 and IRIS, we use the decoders provided within the respective algorithms, both of which are CNN-based decoders.
>
> We have included the number of parameters for the decoder in each baseline in the Table below. For DreamerV3 and IRIS, we follow the default parameters from the original implementations. In response to the reviewer’s suggestion, we have also matched the decoder sizes for DreamerV3 and IRIS (denoted as DreamerV3 Large and IRIS Large). The decoded open-loop rollout images are shown in [Figure 4](https://tinyurl.com/mrku8uza). We observe that DreamerV3 Large shows improved prediction quality compared to the original DreamerV3. However, IRIS Large demonstrates slightly worse performance than IRIS, which could be due to the fact that IRIS requires identical parameters for both its encoder and decoder, making the model more difficult to optimize.
>
> | Model              | Decoder Size  |
> |--------------------|--------------|
> | Ours              | 10,140,163    |
> | IRIS              | 1,821,827     |
> | DreamerV3         | 6,985,667     |
> | IRIS Large        | 11,126,979    |
> | DreamerV3 Large   | 11,256,577    |
>
>
>
> [1] LIBERO: Benchmarking Knowledge Transfer for Lifelong Robot Learning
>
> [2] Diffusion Policy: Visuomotor Policy Learning via Action Diffusion

---

### Official Review · Reviewer_4KWu · 2025-03-13

**Overall Recommendation:** 4

**Summary:**

The paper introduces DINO-WM, a world model that operates within the DINOv2 representation space without the need for reconstruction. The model is trained using teacher forcing with a frame-level causal mask. Due to its task-agnostic nature, DINO-WM can be used for zero-shot model predictive control without demonstration collection, reward modeling, or learning inverse dynamics models.

**Claims And Evidence:**

While leveraging DINOv2 representations as world states achieves superior results compared to image reconstruction, its working mechanism is unclear. The authors claim that reconstruction-based methods contain insufficient task information, but they do not provide evidence that the frozen DINOv2 representations are better in task representation.

**Essential References Not Discussed:**

As far as I know, all closely related works are cited appropriately.

**Experimental Designs Or Analyses:**

The paper provides extensive experiments with sufficient details.

**Methods And Evaluation Criteria:**

The proposed methods are aligned with the motivation and evaluated using appropriate criteria.

**Other Comments Or Suggestions:**

I believe this paper presents a solid practice in world model design and model predictive control. The generality of the proposed method could inspire future research a lot.

**Other Strengths And Weaknesses:**

W1) **Latent world model is not new.** Many related works have adopted the latent world model design, and switching to DINOv2 representation space is somewhat limited in technical innovation. However, the extensive experiments still make this work an insightful contribution to the community.

W2) **The reasons why DINOv2 performs better are not clearly analyzed.** As I mentioned earlier, the authors fail to provide evidence that the frozen DINOv2 representations are better in task representation. In addition, the authors only compare backbones with global representations in Tables 2-4, which are inherently unsuitable for capturing spatial relationships. It would be more convincing if other semantic-rich alternatives, such as MAE [1], are included in the study.

[1] Masked Autoencoders Are Scalable Vision Learners. Kaiming He, et al.

**Questions For Authors:**

Q1) **DINOv2 input resolution.** If I remember it correctly, the pretrained DINOv2 backbone accepts images of 224x224 pixels as input. Therefore, I am curious why the authors resize the input images to 196x196 pixels (Line 927). Does it still work well without finetuning?

**Relation To Broader Scientific Literature:**

The paper introduces DINOv2 as the representation space of the world model, thereby alleviating the need for image reconstruction both during training and testing. While this concept of latent world model is not new, the proposed world model demonstrates superior performance and could serve as a solid baseline for future research.

**Theoretical Claims:**

I have checked the theoretical claims in this paper.

---

> ### Author Rebuttal · Authors · 2025-04-01
>
> We thank the reviewer for their constructive feedback, and for acknowledging that DINO-WM “demonstrates superior performance and could serve as a solid baseline for future research” and “could inspire future research a lot.” We address the issues raised in the review below.
>
> >**“why DINOv2 performs better are not clearly analyzed”“they do not provide evidence that the frozen DINOv2 representations are better in task representation”**
>
> While our manuscript presents results on feature reconstruction and task planning, demonstrating that DINOv2 patch features enable more accurate world modeling, we now provide further analysis of the features themselves. A well-established method for evaluating feature quality in downstream control tasks is linear probing, which assesses how well the features encode task-relevant state information. We conducted linear probe experiments on PointMaze, PushT, and Wall, comparing DINO-S Patch, DINO-S CLS, DINO-B Patch, IR3M, and Pre-trained MAE (DINO-S and DINO-B denotes DINO model with ViT Small and Base architecture, respectively). The validation loss for these linear probes is reported in [Table 2](https://tinyurl.com/mpurtfs), where DINO-S Patch and DINO-B Patch achieve the lowest validation loss, indicating their superior task representation capabilities.
>
> To further analyze the DINOv2 features, we visualize the principal components after performing PCA on the patch embeddings from DINO-S on our deformable environments which has the most complex state space, following a procedure similar to that in the original DINOv2 paper ([Figure 2](https://tinyurl.com/r39zsb4r)). The visualizations show that DINOv2 features effectively identify objects of interest, distinguish the agent, and separate the foreground from the background, reinforcing that frozen DINOv2 representations encode task-relevant information effectively.
>
> >**“the authors only compare backbones with global representations in Tables 2-4, which are inherently unsuitable for capturing spatial relationships”“It would be more convincing if other semantic-rich alternatives, such as MAE [1], are included in the study”**
>
> We totally agree that having more baselines with semantic-rich baselines would provide further insights. We first note that for the IRIS baseline which we compare with in Table 1 and Table 3 of the manuscript, it encodes an image with 16 tokens instead of a global representation, which is also capable of capturing spatial relationships explicitly.
>
> We ran a new experiment on PushT using a pre-trained MAE suggested by the reviewer. The performance is presented in the Table below. We observe that the WM trained with MAE features has lower final MPC performance than the original DINO-WM. We hypothesize this is because MAE prioritizes reconstruction over task relevance, as indicated by linear probe results. Additionally, even the smallest MAE encoder, which we used in the experiments, is still significantly larger than the DINO-S model, with feature dimensionality twice than that of DINO-S. This highlights that a larger and more expensive feature extractor does not necessarily translate to better task performance, and the higher computational cost of MAE makes it a less efficient choice.
>
> | Encoder Model | Encoder Param Count | Feature Size | MPC  |
> |--------------|---------------------|--------------|------|
> | DINOv2      | 22,056,576           | 384          | 0.90 |
> | MAE         | 85,798,656           | 768          | 0.86 |
>
> >**“Q1 DINOv2 input resolution.”**
>
> Although DINOv2 is pre-trained on 224×224 images, its ViT-based architecture processes images as fixed-size patches, allowing it to handle arbitrary image sizes as long as the height and width are divisible by the patch size (e.g., 14 for DINO-S). In fact, the original DINOv2 paper [1] explicitly demonstrates its ability to work with non-224×224 images, including rectangular and high-resolution images (Section 7.5).
>
> In DINO-WM, our choice of 196×196 input resolution is purely an engineering decision. Our decoder assumes a 16x spatial upscaling, and we aimed to match the environment’s 224×224 observation shape after decoding. Since DINOv2’s fixed patch size is 14, an input of 196×196 results in a 14×14 patch grid, which decodes to 224×224 after the 16× upscaling. However, using 224×224 directly with an additional interpolation layer at the decoder output would be equally valid.
>
> To verify this, we conducted an ablation on PointMaze with input sizes ranging from 28×28 to 280×280. The results in [Table 3](https://tinyurl.com/ycyua34p) show that image size 224×224 performs on par with 196×196. This shows that our choice does not limit DINOv2’s capabilities. Moreover, the ability of DINO-WM to process images of various sizes enhances its flexibility in modeling environments of varying complexity, allowing for adaptive trade-offs between granularity and computational efficiency.
>
>  [1] DINOv2: Learning Robust Visual Features without Supervision

---

> > ### Comment · Reviewer_4KWu · 2025-04-03
> >
> > Thank the authors for their comprehensive response. The extensive experiments have addressed my concerns. I will increase my rating to 4 and hope the authors will include the results in the revision.

---

> > > ### Author Response · Authors · 2025-04-05
> > >
> > > Thank you for your thoughtful review and for increasing your score. We're glad the additional experiments addressed your concerns, and we will include the results in the revised manuscript.

---

### Official Review · Reviewer_57WD · 2025-03-13

**Overall Recommendation:** 4

**Summary:**

This paper proposes DINO-WM, a task-agnostic world model that predicts future visual features using DINOv2 embeddings instead of reconstructing raw observations. Trained on offline trajectories with a Vision Transformer, it enables zero-shot test-time optimization via model predictive control. DINO-WM outperforms prior methods in goal-reaching success (45% improvement) and world modeling quality (56% improvement) across diverse tasks like maze navigation and robotic manipulation. By leveraging high-level feature prediction, it achieves flexible planning without task-specific retraining or auxiliary supervision.

**Claims And Evidence:**

DINO-WM claims to produce high-quality world modeling, supported by a 56% improvement in LPIPS over prior methods. It claims to achieve high success in reaching arbitrary goals, with a 45% improvement over previous approaches. Additionally, it claims to generalize across task variations, such as different maze layouts and object shapes, outperforming prior work in diverse environments.

**Essential References Not Discussed:**

No.

**Experimental Designs Or Analyses:**

Yes. The experiments are well-structured to evaluate DINO-WM’s ability to learn from offline datasets, optimize behavior at test time, and generalize across tasks. Comparisons against state-of-the-art baselines show that DINO-WM significantly improves world modeling quality. Results confirm that using DINOv2 patch embeddings enhances planning performance.

**Methods And Evaluation Criteria:**

Yes

**Other Comments Or Suggestions:**

No.

**Other Strengths And Weaknesses:**

Most of the experiments focus on variations of reaching tasks (e.g., non-prehensile manipulation), making it unclear whether the same performance will hold for contact-rich manipulation. Specifically, it is uncertain whether frozen DINOv2 feature patches provide sufficient resolution for contact-rich tasks (even just for pick-and-place tasks). Additionally, for dexterous manipulation with high-DOF action spaces, it is unclear whether planning will be fast enough to find a solution, given that planar pushing already takes ~50 seconds planning time. Lastly, in cluttered environments with complex backgrounds, the model's performance remains uncertain.

**Questions For Authors:**

1. Can you show examples of pick-and-place tasks, preferably with a non-trivial SE(3) action space beyond simple table-top pick-and-place?
2. Can you demonstrate your existing experiments in environments with cluttered backgrounds?
3. Does increasing the latent-space planning horizon degrade performance?
4. How does changing the patch size impact performance?
5. Figure 4 is very compelling, but was your decoder trained on the same data as DreamerV3 and IRIS?

**Relation To Broader Scientific Literature:**

Prior world modeling work learns the dynamics model together with the embedding function. This work leverages frozen DINO features and learns the dynamics model in isolation, showing that a good embedding space makes world modeling + MPC a winning recipe.

**Theoretical Claims:**

No theoretical claims.

---

> ### Author Rebuttal · Authors · 2025-04-01
>
> We thank the reviewer for their constructive feedback, for identifying that DINO-WM “significantly improves world modeling quality” and shows “a good embedding space makes world modeling + MPC a winning recipe”. We address the issues raised in the review below.
>
> >**“it is uncertain whether frozen DINOv2 feature patches provide sufficient resolution for contact-rich tasks”**
>
> We show that frozen DINOv2 patch features capture state accurately, even in contact-rich tasks like PushT and deformable manipulation. Figure 4 in the manuscript illustrates how they precisely represent particle positions—challenging for global features like ImageNet-pretrained ResNet or R3M.
>
> We further trained a DINO-WM on LIBERO [1], a tabletop environment manipulating diverse objects, with third-person image observations and a 7DoF action space. [Figure 1](https://tinyurl.com/5eckpz75) compares open-loop rollouts of WMs trained with DINO patch features vs. DINO CLS features. Reconstruction scores for the predicted frames can be seen in [Table 7](https://tinyurl.com/y4thskrv). This shows that DINO patch features can accurately represent the object within the gripper, whereas global representations struggle with dynamic elements—both the object in the gripper and the gripper itself. This further reinforces the suitability of patch features for contact-rich interactions.
>
> In our response to **Reviewer 4KWu**, we provide linear probe results on the environment state using DINO patch features, along with PCA visualizations, demonstrating its ability to capture task-relevant information.
>
> >**“it is unclear whether planning will be fast enough”**
>
> We address planning efficiency in 3 ways.
> 1. We improved our inference code for DINO-WM since submission. For the same hyperparameters, planning now takes 15.89s, compared to 53s reported in the manuscript.
> 2.  While we report results for CEM with 100 samples per iteration, this can be adjusted based on task complexity. [Table 1](https://tinyurl.com/2mn6aben) presents the tradeoff between sample size, planning time, and performance on PushT. This flexibility allows us to balance computational efficiency with performance depending on task requirements.
> 3. Training and planning with a DINO-WM using a larger frameskip can speed up planning. This effectively increases the planning horizon as each prediction covers a longer time span. We train a DINO-WM with frameskip 25 on PushT and report the performance in [Table 6](https://tinyurl.com/525kjsza). While modeling long-term dependencies is more challenging as frameskip increases, it presents an opportunity to improve efficiency, making hierarchical planning promising directions for future research.
>
> >**“In cluttered environments with complex backgrounds, the model's performance remains uncertain”**
>
> We ran experiments on ClutteredPushT, where the background of the PushT environment is a cluttered real-world tabletop. Open-loop rollouts of a trained DINO-WM is in [Figure 3](https://tinyurl.com/49j2c5av), and we report the task performance in [Table 5](https://tinyurl.com/4wvxtw84) compared to the original PushT.
>
> In ClutteredPushT, DINO-WM can still identify the effect of agent actions and predict the motion of relevant objects. The final planning performance is only marginally degraded. This shows DINO-WM’s capability of modeling environments with complex backgrounds.
>
> Additionally, we have trained an unconditional world model on the CLEVRER [2] dataset where multiple objects may collide. Videos of open-loop rollouts are provided [here](https://tinyurl.com/cx59vz8p).
>
> **Q1. Environment with SE(3) action space.**
>
> We further provide results of DINO-WM on LIBERO which has a 7DoF action space. Due to space limit, we refer to our response to **Reviewer dmt6** for planning videos and performance.
>
> **Q2.** Addressed in the ClutteredPushT experiment.
>
> **Q3. Does increasing the latent-space planning horizon degrade performance?**
>
> Not necessarily. Longer planning horizon enables discovering states that are further away, with the tradeoff that the WM’s long-term prediction would be less accurate. We balance this via planning with receding horizons as in our deformable environments.
>
> **Q4.** We conduct ablations with different image sizes (yielding different patch sizes after DINO encoder) on PointMaze. We report the planning success rate (SR) for CEM, MPC, and the predicted frame’s image scores in  [Table 3](https://tinyurl.com/ycyua34p).
> With MPC, all models eventually obtain a decent SR, but models with larger patch size are able to achieve better SR with CEM. This shows that bigger patch sizes can contain more precise state information, making the world model more accurate for zero-shot open-loop planning.
>
> **Q5.** Yes, our decoder is trained on the same data with all baselines including DreamerV3 and IRIS.
>
> [1] LIBERO: Benchmarking Knowledge Transfer for Lifelong Robot Learning
>
> [2] CLEVRER: CoLlision Events for Video REpresentation and Reasoning

---

### Decision · Program_Chairs · 2025-05-01

**Decision:**

Accept (poster)

**Comment:**

## Summary

This paper introduces DINO-WM, a novel approach to world modeling that leverages pre-trained DINOv2 visual features instead of reconstructing raw observations. The key innovation is using patch-level features from DINOv2 and learning a dynamics model in this rich representation space, which enables high-quality predictions and effective test-time planning. DINO-WM is trained on offline trajectories and can perform zero-shot planning for arbitrary goals without needing expert demonstrations, reward functions, or inverse dynamics models.

## Review Process

The paper received mixed initial reviews, with scores ranging from 2 to 4. Given this divergence in reviewer opinions, I initiated a discussion to better understand the strengths and limitations of the work. The authors provided thorough responses during rebuttal, addressing concerns about planning efficiency, performance on complex environments, and additional experiments with different foundation models. Following the discussion and rebuttal, one reviewer increased their score, resulting in three "Accept" ratings and one "Weak accept" rating.

## Strengths

- **Representation Innovation**: DINO-WM demonstrates that frozen pre-trained visual features can serve as an effective state space for world modeling, outperforming models that learn representations from scratch.
- **Strong Performance**: The approach shows substantial improvements over prior methods, with a 56% boost in world modeling quality (LPIPS) and 45% higher goal-reaching success rates across diverse environments.
- **Generalizability**: DINO-WM generalizes well to novel environment configurations, such as different maze layouts and object shapes.
- **Zero-shot Planning**: The method enables planning for arbitrary goals without requiring task-specific training, expert demonstrations, or reward signals.

## Limitations

- **Planning Efficiency**: While the authors improved planning speed from 53s to 15.89s in their rebuttal, real-world deployment may still require further optimization.
- **Dataset Requirements**: DINO-WM needs comprehensive offline datasets with sufficient state-action coverage, which could be challenging to collect in some real-world settings.
- **Complexity of Evaluated Tasks**: Most experiments focus on relatively simple simulation environments, though the authors provided additional results on more complex environments (LIBERO, ClutteredPushT) in their rebuttal.
- **Foundation Model Analysis**: While the authors evaluated different models in response to reviewer requests (MAE, SigLIP+DINO), a more comprehensive analysis of why DINOv2 features work particularly well would strengthen the paper.

## Recommendation

After carefully considering the reviews, author responses, and subsequent discussion, I recommend acceptance of this paper. DINO-WM represents a significant advance in world modeling by effectively leveraging pre-trained visual features to enable zero-shot planning. The authors have addressed key concerns raised during review, demonstrating the approach's effectiveness on more complex tasks and providing a thorough analysis of design choices.

The discussion among reviewers highlighted that despite some limitations, the core contribution of using pre-trained patch-level features for world modeling is valuable and opens promising directions for future research. I encourage the authors to incorporate their rebuttal material, especially the additional experiments and analyses, into the final version of the paper.